# Two energy barriers and a transient intermediate state determine the unfolding and folding dynamics of cold shock protein

Haiyan Hong[1,4], Zilong Guo[1,2,3,4], Hao Sun[1], Ping Yu[1], Huanhuan Su[1], Xuening Ma[1] & Hu Chen [1,2,3✉]

Cold shock protein (Csp) is a typical two-state folding model protein which has been widely studied by biochemistry and single molecule techniques. Recently two-state property of Csp was confirmed by atomic force microscopy (AFM) through direct pulling measurement, while several long-lifetime intermediate states were found by force-clamp AFM. We systematically studied force-dependent folding and unfolding dynamics of Csp using magnetic tweezers with intrinsic constant force capability. Here we report that Csp mostly folds and unfolds with a single step over force range from 5 pN to 50 pN, and the unfolding rates show different force sensitivities at forces below and above ~8 pN, which determines a free energy landscape with two barriers and a transient intermediate state between them along one transition pathway. Our results provide a new insight on protein folding mechanism of two-state proteins.

[1] Research Institute for Biomimetics and Soft Matter, Fujian Provincial Key Lab for Soft Functional Materials Research, Department of Physics, Xiamen University, Xiamen 361005, China. [2] Center of Biomedical Physics, Wenzhou Institute, University of Chinese Academy of Sciences, Wenzhou 325000, China. [3] Oujiang Laboratory, Wenzhou, Zhejiang 325000, China. [4] These authors contributed equally: Haiyan Hong, Zilong Guo. ✉email: chenhu@xmu.edu.cn

Most small single-domain proteins fold to a unique three-dimensional structure, and the thermodynamics of protein folding and unfolding can be described by a two-state model with a highly populated native state and unfolded state only[1,2]. But the detailed mechanism of protein folding/unfolding, such as transition pathways, properties of the transition state, and the existence of intermediate states, needs further study by newly developed highly sensitive techniques. Theoretically, a funnel-shaped free energy landscape with a native state at the bottom provides a general picture of proteins folding[3,4]. Transition state separates native state and unfolded state by an energy barrier. Intermediate states are local traps on the free energy landscape. The transition pathway connects unfolded state, native state, and possible intermediate states through transition states between them[5].

Single molecular manipulation techniques apply force to tilt the free energy landscape and record the conformation transition process of a single protein through the extension and force signals[6,7]. Force-dependent transition rate and detection of short-lived intermediate states provide hints to study the complex free energy landscape of proteins[8]. Cold shock protein B (Csp) from *Thermotoga maritima* is a model protein to study protein folding dynamics with a single peptide chain of 66 amino acids[9–11]. Csp has a highly conserved single-domain structure, which can interact with single-stranded DNA or RNA[12–14]. Csp functions as nucleic acid chaperons to prevent the formation of DNA/RNA secondary structures at low temperature[15]. Previous bulk experiments and a single molecular fluorescence experiment both suggested that Csp is a typical two-state model protein[16–18]. Single-molecule force spectroscopy by atomic force microscope (AFM) measured force-induced unfolding of Csp at constant pulling speeds from 100 to 2000 nm s$^{-1}$. The single-step unfolding signal of Csp is consistent with the two-state model, and pulling-speed-dependent unfolding forces gave unfolding distance $x_u = 0.49$ nm[19]. At this pulling-speed range, the average unfolding force is between 75 and 90 pN, and the observed unfolding events happen at forces from ~50 to ~150 pN. Therefore, under mechanical conditions of both zero force and high force, Csp shows two-state behavior.

Recently, force-clamp AFM was used to stretch Csp and record extension time course[20]. Csp was found to unfold with about 50% probability through multiple heterogeneous pathways with long-lived intermediate states (lifetime of seconds) at a moderate force range from 40 to 60 pN. Unfolding steps with a size smaller than the expected unfolding step of whole Csp provides direct evidence of a stable intermediate state during Csp unfolding process. Molecular dynamic simulation at 200 pN gave unfolding pathways with an intermediate state whose lifetime is from several nanoseconds (ns) to tens of ns[20–22]. Additional simulation based on a coarse-grained model also found some intermediate states with a lifetime of several ns at low forces of 5–20 pN[23]. Though the simulations support the existence of unfolding intermediate states, the lifetime of the intermediate states cannot be compared directly between experiments and simulations. Single-molecule manipulation techniques like AFM and optical tweezers connect an effective Hooke spring to the tethered biomolecule. Therefore, a realization of constant force relies on a feedback system. As feedback system usually has finite response time, it might lead to errors in the measurement of fast dynamic process[24]. Compared to the force feedback system, passive force-clamp mode and constant-trap-position mode are more recommended in optical tweezers[25].

Magnetic tweezers can maintain stable constant force over a long time without any sophisticated force feedback system[26–28]. Here, we report the full-scale study of force-dependent folding and unfolding dynamics of Csp under constant forces using magnetic tweezers. At forces close to its critical force of about 6.6 ± 0.2 pN, equilibrium folding and unfolding dynamics give folding free energy of 12.6 ± 0.4 $k_BT$. At forces up to 50 pN, unfolding rates were measured by force-jump experiments. We did not observe an intermediate state during both folding and unfolding processes if the measurement was done within 3 h after sample preparation. We found that the unfolding distance $x_u$ at forces smaller than 8 pN is much larger than that at large forces. A free energy landscape with two barriers and a transient intermediate state is constructed based on the experimental results.

## Results

**Force-dependent unfolding of Csp measured at constant loading rates**. We tethered the protein of Csp with I27$_2$ on both flanks through a C-terminus SpyTag attached to the glass surface coated with SpyCatcher and N-terminus biotin attached to a streptavidin-coated paramagnetic bead M270 (Fig. 1a)[29]. To determine the extension trajectory of Csp during the force-dependent unfolding and folding processes, we increased force from ~1.2 to ~25 pN at a loading rate of 1 pN s$^{-1}$. An unfolding event was observed at ~16.6 pN with a single step of ~16.5 nm. After arriving at 25 pN, we decreased force to 1.2 pN with a loading rate of $-1$ pN s$^{-1}$, and a refolding step was recorded at ~4.7 pN with a single step of ~8.7 nm (Fig. 1b). The tether was held for 30 s to make sure the protein folds to its original native state. Then force was increased at the same loading rate to 80 pN. Besides the unfolding step of Csp, four unfolding steps of I27 with step size ~25 nm were observed at a force of ~80 pN, which confirms that a single target protein was stretched. In force ramp experiments, almost all Csp unfolding events present a single step without long-lifetime intermediates, which is consistent with a standard two-state model.

To get enough statistical data of Csp unfolding, multiple stretching cycles were done at the same loading rate. Unfolding force distribution $P(f)$ at a loading rate of 0.4 pN s$^{-1}$ was obtained from 81 unfolding events. The average unfolding force is about ~13.4 pN which is close to the peak force of this distribution. We suppose that force-dependent unfolding rates follow Bell's model: $k_u(f) = k_u^0 \exp(fx_u/k_BT)$, where $k_u^0$ denotes the unfolding rate at zero force if the formula can be extrapolated to zero force, $x_u$ the unfolding distance, $k_B$ the Boltzmann constant, and $T$ the absolute temperature[30]. Fitting of the obtained unfolding force distribution with Eq. (2) gives $k_u^0 = 0.005$ s$^{-1}$ and $x_u = 0.97$ nm (Fig. 1c) which is bigger than the result of 0.49 nm from AFM experiment[19,31]. The corresponding folding force distribution at a loading rate of $-0.4$ pN s$^{-1}$ was obtained from 66 folding events. The average folding force is about ~6.1 pN which is also close to the peak force of this distribution. Fitting of the obtained folding force distribution with Eq. (3) based on Bell's model gives the folding rate at zero force $k_f^0 = 1.3 \times 10^7$ s$^{-1}$ and the folding distance $x_f = 11.1$ nm (Fig. 1d)[32,33]. This $k_f^0$ is even faster than the folding speed limit, which is clearly an unreasonable result of an improper model. To obtain a more accurate force-dependent unfolding property of Csp, we stretched protein Csp at different loading rates from 0.4 to 16 pN s$^{-1}$ (Supplementary Fig. 1). The average unfolding force is not a linear function of the logarithm of loading rate (Fig. 1e). Fitting with Eq. (4) gives $k_u^0 = 0.002$ s$^{-1}$ and $x_u = 1.42$ nm at loading rates from 0.4 to 4 pN s$^{-1}$, and gives $k_u^0 = 0.03$ s$^{-1}$ and $x_u = 0.64$ nm at loading rates from 4 to 16 pN s$^{-1}$.

The obtained fitting parameters, especially $x_u$, show a large discrepancy between the above two measurements. Additionally, the obtained $x_u$ at low force range is bigger than previous results by AFM[19]. Therefore, we suspect that force-dependent unfolding and folding rates cannot be described by Bell's model.

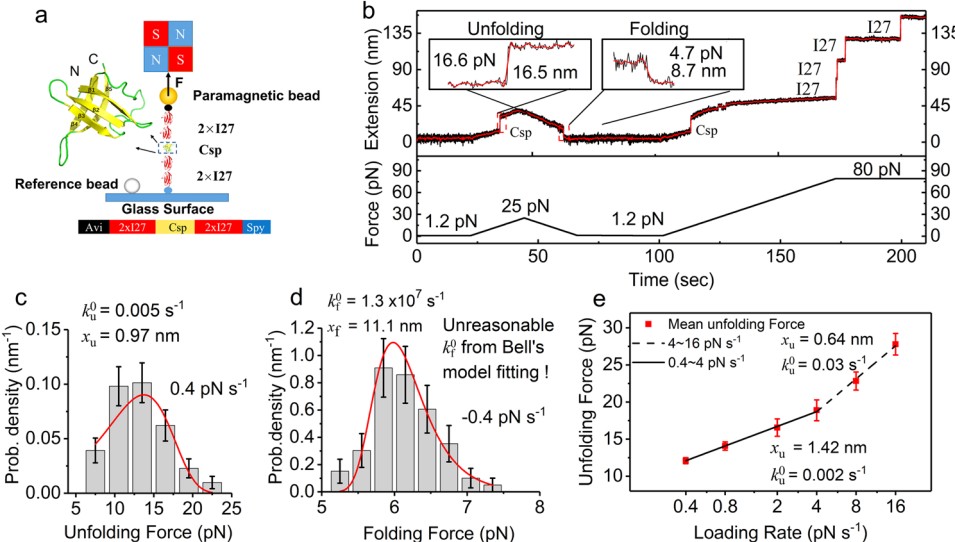

**Fig. 1 Force-dependent unfolding and refolding of Csp measured by magnetic tweezers at constant loading rates. a** Sketch of magnetic tweezers pulling protein construct of 2xI27-Csp-2xI27 with the AviTag at N-terminus attached to a streptavidin-coated paramagnetic bead and SpyTag at C-terminus attached to the glass surface coated with SpyCatcher. Zoomed in figure shows a structure of Csp. **b** Extension time course of 2xI27-Csp-2xI27 in a force cycle at a loading rate of ±1 pN s$^{-1}$ from 1.2 to 25 pN, an unfolding step and a refolding step were observed at ~16.6 and ~4.7 pN, respectively. After waiting for 30 s at 1.2 pN, a force was increased at a loading rate of 1 pN s$^{-1}$ to 80 pN, additional four unfolding steps I27 were observed after unfolding of Csp. Raw data were recorded at 200 Hz (black) and smoothed in a time window of 1 s (red). **c** Unfolding force distribution at a constant loading rate of 0.4 pN s$^{-1}$ were obtained from 81 unfolding events. The fitting of unfolding probability density with Eq. (2) gives a $k_u^0 = 0.005$ s$^{-1}$ and $x_u = 0.97$ nm. **d** Folding force distribution at a constant loading rate of $-0.4$ pN s$^{-1}$ were obtained from 66 folding events. The fitting of folding probability density with Eq. (3) gives a $k_f^0 = 1.3 \times 10^7$ s$^{-1}$ and $x_f = 11.1$ nm. **e** The average unfolding forces were obtained at a loading rate from 0.4 to 16 pN s$^{-1}$ from five different tethers. Linear fitting with Eq. (4) gives $k_u^0 = 0.002$ s$^{-1}$ and $x_u = 1.42$ nm at loading rates from 0.4 to 4 pN s$^{-1}$ (black line) and gives $k_u^0 = 0.03$ s$^{-1}$ and $x_u = 0.64$ nm at loading rates from 4 to 16 pN s$^{-1}$ (black dashed line). The number of unfolding events at different loading rates are from 34 to 120. Error bar shows the standard error of the mean.

Theoretically, the model-independent unfolding rate can be obtained from unfolding force distribution $P(f)$, but it requires a large amount of data to give an accurate $P(f)$[34].

**Constant force measurement.** Taking advantage of the intrinsic constant-force capability of magnetic tweezers, we directly measure the force response of Csp to constant forces. Based on previous constant loading rate measurement, we know the unfolding force of Csp is about 16 pN and the folding occurs at about 5 pN. The critical force at which Csp folds and unfolds with the same rates must be between these two forces. We found that both folding and unfolding transitions of Csp can be observed at constant forces from 5 to 7.5 pN (Supplementary Fig. 2). The extension time courses at 5.5, 6.3, and 7 pN are shown in Fig. 2a as examples. The right panel shows the Gaussian fitting of the relative frequency histogram of the smoothed extension with two peaks corresponding to the native and the unfolded states of Csp, respectively. We did the equilibrium measurements for about 1 h, and no intermediate state with an extension between native state and unfolded state was found. Unfolding and folding probability as a function of time are obtained from the cumulative distribution of lifetime of the native state and unfolded state, respectively. The exponential fitting gives corresponding $k_u$ and $k_f$ at each force (Fig. 2b, c). Folding and unfolding probability of Csp at different constant forces are obtained from the bimodal Gaussian fitting of extension (Fig. 2d), which can be fitted with Eq. (9). Four independent measurements give critical force $f_c = 6.6 \pm 0.2$ pN and $\Delta x = 14 \pm 1$ nm which is consistent with the recorded step size (Supplementary Fig. 3).

In order to explore the unfolding rate at a higher force range, we did the force-jump experiment from 10 to 50 pN (Fig. 3 and Supplementary Fig. 4). Firstly, we hold the protein at force 1 pN and changed force to various high values abruptly to record the unfolding step of Csp. After observing the unfolding step of Csp, we decreased the force to 1 pN and kept a force of 1 pN for 20 sec to make Csp fold to the native state. Then another cycle of force-jump experiment can be done. Figure 3 shows one cycle at each force, and the upper traces show zoom-in of unfolding events at each force. The unfolding of Csp happened in one single step, and no intermediate state was detected if the experiment was finished in 3 h. The signal after Csp unfolding at 30 pN is the unzipping/zipping dynamics of N-SpyTag/SpyCatcher in the tether. This signal can serve as the fingerprint to identify the correct protein tether[35]. From waiting time before unfolding at each force, unfolding rates are determined with the same exponential fitting method shown in Fig. 2b.

The average force-dependent unfolding rate and folding rate obtained from both equilibrium measurement and force-jump measurement are summarized in Fig. 4a, b (also shown in Supplementary Fig. 5). We found that the force-dependent unfolding rate from 5 to 50 pN cannot be described by Bell's model. The data points of unfolding rates are roughly on two linear lines with distinct slopes. The average unfolding rates from 10 to 50 pN can be fitted with Bell's model with fitting parameters $k_{u,1}^0 = (3.2 \pm 0.7) \times 10^{-2}$ s$^{-1}$ and $x_{u,1} = 0.55 \pm 0.03$ nm, which is consistent to the result from previous AFM experiment[19] and biochemistry study[16] which rely on linear extrapolation from unfolding rates measured at high force or high concentration of denaturant conditions, respectively. Fitting of unfolding rate from 5 to 7 pN provides a zero-force unfolding rate $k_{u,2}^0 = (2.0 \pm 1.9) \times 10^{-4}$ s$^{-1}$ which is more than two orders of magnitude smaller than $k_{u,1}^0$, and different unfolding distance $x_{u,2} = 3.1 \pm 0.2$ nm which is about five times bigger than $x_{u,1}$.

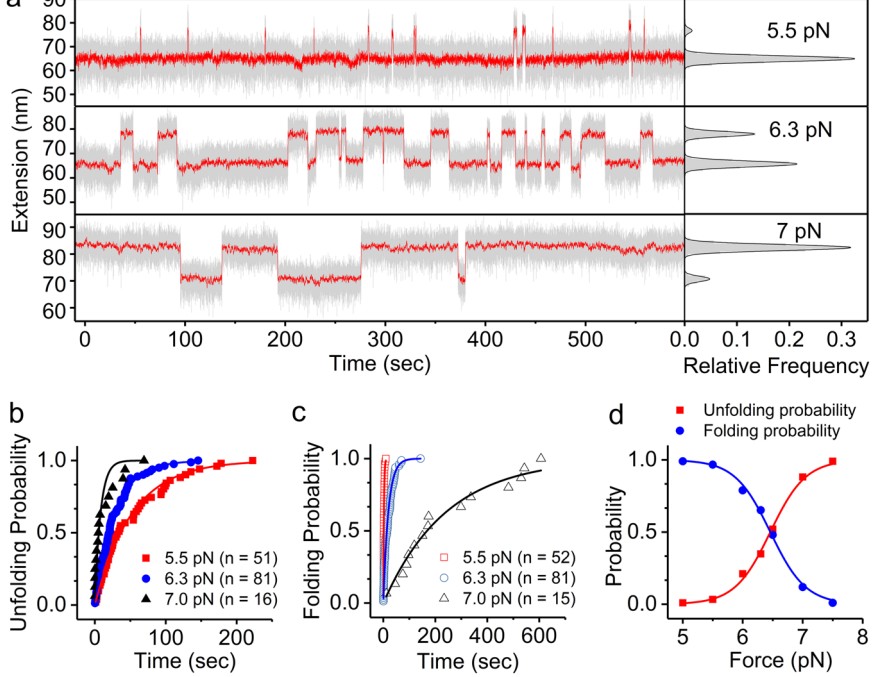

**Fig. 2 Equilibrium measurement of Csp at various constant forces. a** Extension time course of Csp at three constant forces of 5.5, 6.3, and 7 pN were recorded for 600 s. Corresponding relative frequency distributions of the extensions were shown in the right panel and fitted by Gauss functions. Raw data were recorded at 200 Hz (gray) and smoothed over a 5-s time window (red). **b, c** The Unfolding and folding probability of Csp at the three constant forces. Solid lines show exponential fitting curves to determine $k_u$ and $k_f$ of Csp. **d** The force-dependent folding/unfolding probability data points were fitted with Eq. (9).

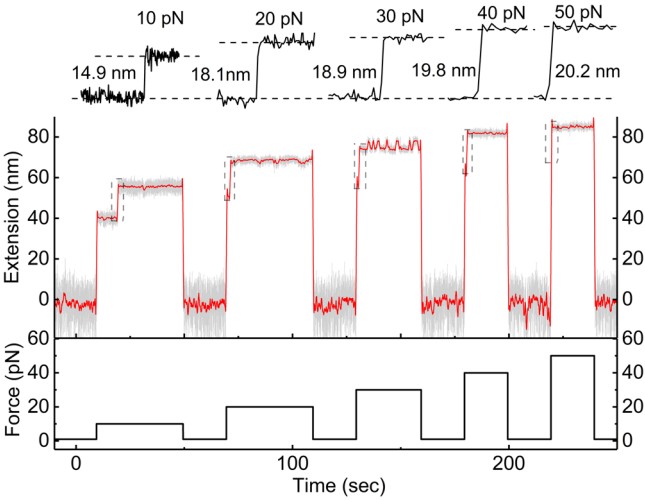

**Fig. 3 Unfolding process of Csp at various constant forces measured in a force-jump experiment.** Unfolding steps were recorded after force jumped from 1 pN to various high forces (10, 20, 30, 40, and 50 pN) on the same protein tether. Before each measurement, a force of 1 pN was applied for 20 sec to make Csp protein folds to its native state. The upper traces show zoom-in of unfolding events at each force. The signal after Csp unfolding at 30 pN is the unzipping/zipping dynamics of N-SpyTag/SpyCatcher in the tether. The red curve shows smoothed extension over a 1-s time window.

As unfolding rates in the force range of 10–50 pN and 5–7 pN are obtained from the force-jump experiment and equilibrium measurement, respectively, we suspect that the different slopes of fitting lines might be due to different methods of measurement. Then we did force-jump measurement from 5 to 50 pN and obtained similar results, which proves that the measured

unfolding rates are not affected by different experimental strategies (Supplementary Fig. 6).

The folding rate obtained from the equilibrium measurement decreases as force increases. Compared to the unfolding rate, the folding rate is much more sensitive to force. The measured force-extension curves of unfolded Csp can be well described by a worm-like chain (WLC) model with a contour length of $L = 26.0 \pm 0.3$ nm and a persistence length of $A = 0.80 \pm 0.04$ nm (Supplementary Fig. 7 and Eqs. (5–6)). We suppose that the folding transition state is the same as the unfolding transition state in a force range of 5–7 pN, then the folding transition state has a size of $l_0 = 4.4$ nm. The force-dependent folding rates can be fitted with Eq. (10) with a zero-force folding rate $k_f^0 = 400 \pm 100 \, \mathrm{s}^{-1}$ (blue solid curve in Fig. 4a, b), which is consistent with the previous biochemistry study[16].

Based on the force-dependent unfolding and folding rates, we can get the force-dependent folding free energy $\Delta G(f)$ according to Eq. (8). Then we can calculate the zero-force protein folding free energy using Eq. (7). Independent measurements give $\Delta G(0) = 12.6 \pm 0.4 \, k_B T$, which is slightly larger than previous biochemistry measurements ($8.5$–$10.5 \, k_B T$)[9,16].

**Free energy landscape revealed by two different unfolding transition distances.** The distance between N-terminus and C-terminus is 1.32 nm of Csp in the crystal structure. The slopes of the two fitting lines of unfolding rates give distinct unfolding distances $x_{u,1} = 0.55$ nm and $x_{u,2} = 3.1$ nm, which imply that there are two transition states (TS) located at N-C distances of 1.87 nm (for TS1) and 4.42 nm (for TS2), respectively (Fig. 4c). The size of unfolded state can be estimated as ~6 nm based on the free-joint chain model of a polypeptide.

If we set the intrinsic unfolding rate[36] of Csp as ~$10^6 \, \mathrm{s}^{-1}$, then the energy barriers can be estimated to be $17.2 \, k_B T$ (TS1) and

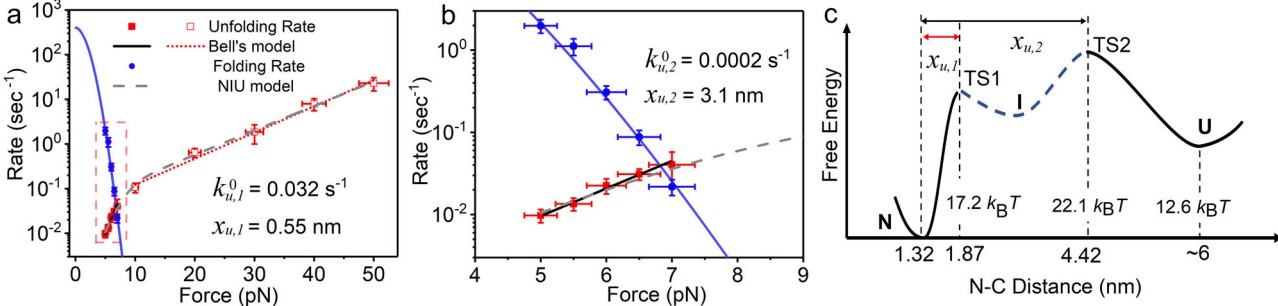

**Fig. 4 Force-dependent folding rates and unfolding rates and free energy landscape of Csp. a** The average unfolding and folding rates of Csp measured by equilibrium constant force experiment (solid symbol) and force-jump experiment (open symbol). **b** is a zoomed-in of the dashed rectangular region in **a**. The unfolding rates were fitted using Bell's model at force range from 5 to 7 pN (black line) and from 10 to 50 pN (red short dotted line) separately, which gives two different $x_u$: $x_{u,1} = 0.55 \pm 0.03$ nm, $x_{u,2} = 3.1 \pm 0.2$ nm, and corresponding parameters $k_{u,1}^0 = (3.2 \pm 0.7) \times 10^{-2} \mathrm{s}^{-1}$, $k_{u,2}^0 = (2.0 \pm 1.9) \times 10^{-4} \mathrm{s}^{-1}$, respectively. The average rates are from four different tethers and the details are shown in Supplementary Fig. 5. Nonlinear force-dependent unfolding rates at force range 5–50 pN are fitted with NIU model (gray dashed line, Supplementary Eq. 1–5). The folding rates were fitted using Arrhenius' law (solid blue line, Eq. (10) and $k_f^0 = 400 \pm 100 \mathrm{s}^{-1}$. Error bar of rate shows the standard error of the mean, and force is estimated to have 5% uncertainty. **c** Free energy landscape of Csp at zero force is plotted as a function of N-C distance. With free energy of N as a reference, the free energy of U is obtained by Eq. (7). Free energy of TS1 (TS2) is calculated from fitting parameters of Bell's mode $k_{u,1}^0$ ($k_{u,2}^0$) based on equation $k_u^0 = k^* \exp(-\Delta G^{\ddagger})$ where the pre-factor $k^* = 10^6 \mathrm{s}^{-1}$. An intermediate state located between TS1 and TS2 might exist but is not stable enough to be recorded in experiments.

$22.1\ k_\mathrm{B}T$ (TS2). Between TS1 and TS2, we speculate that there exists an intermediate state (I), although it was not observed in our experiment directly. With free energy of native state as a reference, a free energy landscape is constructed along reaction coordinate of N-C distance based on folding free energy, barrier heights, and locations of TS1 and TS2 (Fig. 4).

Based on the free energy landscape, we can modify the two-state model by incorporating the intermediate state I between N and U states (NIU model):

$$N \rightleftharpoons I \rightarrow U \qquad (1)$$

We found that the unfolding rate is not sensitive to the location of state I along the unfolding pathway (Fig. 4c). As we cannot detect state I directly in the recorded time course of extension, the state I must be not very stable. We suppose that state I has free energy $\sim 14\ k_\mathrm{B}T$ and is located at a position in the middle of TS1 and TS2, the force-dependent unfolding rates can be fitted with analytical equations of the NIU model derived by Pierse and Dudko[37] (Fig. 4a, Supplementary Note 1, and Supplementary Table 1). Interdependence between parameters is given in Supplementary Fig. 8.

## Discussion

In this study, we measured the folding and unfolding processes of Csp protein by magnetic tweezers beyond the force range of the previous AFM experiment and obtained an unexpected nonlinear force-dependent unfolding rate. In single-molecule manipulation experiments, the dynamic properties of the apparatus, such as the frequency of response, play an important role in the measured results and their interpretation[38,39]. When the molecular transition rates are comparable or faster than the apparatus transition rate[40,41], the measured transition rates can be much slower than the intrinsic molecular rates[42]. Based on the same theoretical framework, we estimated the apparatus transition rate $k_A$ of magnetic tweezers to be about $620\ \mathrm{s}^{-1}$ which is more than one order of magnitude bigger than the fastest unfolding rate of Csp in our measurements (Supplementary Note 2 and Supplementary Fig. 9). Therefore, our results reflect mainly the intrinsic molecular transition rate of Csp.

In summary, force-dependent unfolding and folding rate of Csp at a force range from 5 to 50 pN were measured directly by magnetic tweezers, especially the force-dependent folding rate which is not measured before. The equilibrium measurement at

5–7 pN directly determines the critical force of 6.6 pN and zero-force folding free energy of Csp $\sim 12.6\ k_\mathrm{B}T$.

For most protein tethers, we didn't record an apparent intermediate state in multiple cycles of constant loading rate and force-jump experiment due to the limited five-millisecond time resolution of our magnetic tweezers. This is in contrast to the previous force-clamp AFM measurement which detected several long-lifetime intermediate states during the unfolding process at 40 and 60 pN with more than 50% probability[20]. Occasionally, after several hours of measurement, multistep unfolding events start to be observed (Supplementary Fig. 10). One possibility is that the protein was injured at some unknown position during such a long-time measurement.

We found that the force-dependent unfolding rate cannot be fitted with Bell's model. Unfolding of Csp is more sensitive to force when force is smaller than 8 pN and less sensitive to force at force range greater than 8 pN. A similar phenomenon has been found in GB1 protein[43]. As shown in Fig. 4c, TS1 with $x_{u,1} = 0.55$ nm determines the flexibility of Csp native state. Compared to I27 and GB1, the naive state of Csp is more flexible, which may be the structural basis for its function to interact with single-stranded DNA or RNA as a nucleic acid chaperon[19,44].

Because we cannot directly record the transient intermediate state I between the two transition states TS1 and TS2 from extension steps of folding and unfolding transitions, it indicates that I state must have a relatively high free energy. This intermediate state might be the molten globule state proposed in traditional protein folding theory[45]. Then TS2 with a size of about 4.42 nm serves as the general barrier between the molten globule state and unfolded polypeptide.

From the viewpoint of technology capability, our results demonstrate the importance of measurement over a large force range which is one advantage of magnetic tweezers. If only force-dependent unfolding rates at forces greater than 10 pN are measured as previous AFM measurements[19,20], the transition state of TS2 will be ignored. On the other hand, if only equilibrium folding/unfolding dynamics are measured in the vicinity of critical force 6.6 pN, TS1 cannot be detected. Therefore, measurement of folding and unfolding rates over a large force range is critical to construct the full free energy landscape of proteins, and magnetic tweezers are most suitable to accomplish this measurement among single-molecule manipulation techniques.

## Methods

**Cloning and protein expression.** The recombinant protein construct 6×His-AviTag-I27$_2$-Csp-I27$_2$-SpyTag was made by inserting the DNA sequence of Csp (synthesized by GenScript Biotech) into the vector pET151-I27$_4$, which had two Titin-I27 domains on each side of multiple cloning sites. To generate biotinylated protein, plasmids of pET151-6×His-AviTag-I27$_2$-Csp-I27$_2$-SpyTag and pBirA (Biotin ligase expression plasmid) were transformed into the *E. coli* strain BL21 (DE3). Selected transformants on LB plates were cultured in LB medium (supplemented with chloramphenicol, ampicillin, D-biotin) at 310 K until the optical density (OD) of the bacterial cell reached 0.6. Protein expression was induced with 0.5 mM isopropyl β-D-thiogalactopyranoside (IPTG) for 12 h at 298 K. The cells were harvested by centrifugation and lysed by sonication in a buffer (50 mM Tris, 500 mM NaCl, 10% glycerol, 5 mM imidazole, 5 mM 2-mercaptoethanol, pH 8.0). The target recombinant protein was purified by using Ni-NTA Sefinose (TM) Resin (Sangon Biotech) and Superdex 200 (GE Healthcare,) according to the manufacturer's protocol. The protein was snap-frozen in liquid nitrogen and stored at 193 K.

**Single-molecule magnetic tweezers measurements.** The target protein was coupled between the glass coverslip of a laminar flow chamber and a 2.8 μm diameter paramagnetic bead (Dynabeads M270, Invitrigen) (Fig. 1a). Coverslips were cleaned in an ultrasonic cleaner in 5% Decon90 detergent, then were treated with oxygen plasma cleaner for 10 min. Cleaned coverslips were incubated in a solution of 1% 3-aminopropyltriethoxy-silane (APTES, cat. A3648, Sigma) in methanol for 1 h and rinsed by methanol. Flow chambers were assembled by adding parafilm as a spacer between the APTES-coated coverslip and another cleaned coverslip. Amino functionalized microspheres (Polybead cat.417145, Polysciences) with 3 μm diameter were added and incubated for 20 min in the chamber. They stuck on the surface and were used as a reference to eliminate spatial drift during experiments. Then SpyCatcher protein was immobilized onto the coverslip after 1% Sulfo-SMCC (SE 247420, Thermo Science) was incubated for 20 min in the chamber. The chamber was blocked by 1% BSA in 1xPBS overnight at room temperature. Protein with N-terminus AviTag and C-terminus SpyTag was flowed into chamber and incubated for 25 min, then streptavidin-coated paramagnetic beads M270 were flowed into the chamber to form tether.

Home-made magnetic tweezers were used to apply stretching force to Csp protein tether to study its force-dependent folding and unfolding dynamics. Detailed magnetic tweezers design and force calibration methods refer to our previous publication[27,35]. Constant loading rates between 0.4 and 16 pN s$^{-1}$ were realized by moving the magnets with different speeds. Force-jump experiments were done by moving magnets rapidly in less than 0.15 sec (Supplementary Fig. 11).

**Theoretical model and data analysis.** If we suppose that the unfolding rate follows Bell's model, i.e., $k_u(f) = k_u^0 \exp(f x_u / k_B T)$, the unfolding force distribution $P_u(f)$ at constant loading rate $r$ is:[31]

$$P_u(f) = \frac{k_u^0}{r} \exp\left[\frac{f x_u}{k_B T} + \frac{k_u^0 * k_B T}{r x_u}\left(1 - \exp\frac{f x_u}{k_B T}\right)\right]. \tag{2}$$

Similarly, if the folding rate also follows Bell's model, i.e., $k_f(f) = k_f^0 \exp\left(-f x_f / k_B T\right)$, the folding force distribution $P_f(f)$ is given by:

$$P_f(f) = \frac{k_f^0}{r} \exp\left[\frac{-f x_f}{k_B T} - \frac{k_f^0 * k_B T}{r x_f}\left(\exp\frac{-f x_f}{k_B T}\right)\right]. \tag{3}$$

Equation (2) gives the most probable unfolding force $F_u^*$:[32,33]

$$F_u^* = \frac{k_B T}{x_u} \ln\frac{r x_u}{k_u^0 k_B T}. \tag{4}$$

If we model protein's native state as a solid body, its extension along force direction $x_{Csp}$ changes due to its orientation fluctuation:

$$x_{Csp}(f) = l_0 \coth\left(\frac{f l_0}{k_B T}\right) - \frac{k_B T}{f}, \tag{5}$$

where $l_0$ donates the end-to-end distance of Csp on the basis of the crystal structure of Csp (PDB: 1G6P).

Unfolded peptide is modeled as WLC whose force-extension curve is given by:

$$\frac{f A}{k_B T} = \frac{x_{chain}}{L} + \frac{1}{4\left(1 - \frac{x_{chain}}{L}\right)^2} - \frac{1}{4}, \tag{6}$$

where A denotes the persistence length, $x_{chain}$ the extension, and L the contour length of the peptide. A force-dependent extension $x_{chain}(f)$ is derived by an inverse function of Eq. (6). Force-dependent unfolding and folding step size $\Delta x(f) = x_{chain}(f) - x_{Csp}(f)$. Therefore, protein folding free energy at force $f$ is given by:

$$\Delta G(f) = \Delta G_0 - \int_0^f \Delta x(f') df', \tag{7}$$

where $\Delta G_0$ is folding free energy at zero force. Force-dependent protein folding free energy $\Delta G(f)$ of Csp can also be determined by the ratio between folding rate $k_f(f)$ and unfolding rate $k_u(f)$ at the same force $f$:

$$\Delta G(f) = k_B T \ln\frac{k_f(f)}{k_u(f)}. \tag{8}$$

Therefore, equilibrium measurement gives $\Delta G(f)$ directly, from which $\Delta G_0$ can be determined by Eq. (7).

In the neighboring region of the critical force $f_c$ at which the protein has 50% probability at its native state, $\Delta x$ can be approximately supposed as a constant. Therefore, the probability of protein to be at its native state can be fitted with:

$$P_f(f) = \frac{1}{\exp\left((f - f_c)\Delta x / k_B T\right) + 1}. \tag{9}$$

Folding happens at low forces from a stretched polypeptide to the native state crossing a folding transition state. We assume the folding transition state has extension $x_{TS}(f)$. Therefore, the force-dependent folding rate follows:

$$k_f(f) = k_f^0 \exp\left(-\frac{\int_0^f x_f(f') df'}{k_B T}\right), \tag{10}$$

where $x_f(f) = x_{chain}(f) - x_{TS}(f)$ denotes the force-dependent folding distance.

## Data availability

The data that support the findings of this study are available from the corresponding author upon reasonable request.

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

## Acknowledgements

This work was supported by the National Natural Science Foundation of China (Grant No. 11874309 and 11474237) and 111 project (B16029).

## Author contributions

H.H. and Z.G. designed and did the experiments and analyzed the data, contributed equally to this work. H.S., P.Y., H.S. and X.M. made the flow chambers and some data collection. H.C. directed the project. H.C. and H.H. wrote the paper.

## Competing interests

The authors declare no competing interests.
