## [Peer Review File · Communications Chemistry]

Reviewers' comments:

Reviewer #1 (Remarks to the Author):

The work by Hong et al investigates the (un)folding of Csp from T maritime using magnetic tweezers. The authors investigate the unfolding and folding process over a range of forces and conclude that the protein is a two-state folder, in contrast to previous work using AFM. Magnetic tweezers surely have better resolution than AFM but I find one potential problem that the authors should comment on.

Magnetic tweezers force measurement cannot be calibrated directly, unlike AFM, so it relies on indirect measurements of beads position. Perhaps the most relevant figure to conclude the two-state of Csp is Figure 3, in which there is a clear drift on the baseline force claimed to be 1 pN. Given this low value, a simple inspection reveals a drift of maybe 3-5 pN which completely perturbs the folding process of Csp. So, there is a real chance that the protein is not visiting such a low force to secure proper formation of all the contacts in the protein. Also, why do they see about 20-30 nm previous extension after each pulling?

In addition, they should show the full unfolding of the construct including the I27 modules. Even so, they should make a proper statistic with more than one single molecule, how many traces of single molecules recorded a varying forces do they have? To make a proper assumption about the folding/unfolding a good number of traces must be shown, because even in the AFM there is a chance to see single-event unfolding at different forces, in fact, this is the most probable scenario.

Reviewer #2 (Remarks to the Author):

Dear Editor,

The manuscript "Free energy landscape with two barriers and a transient intermediate state determining the unfolding and folding dynamics of cold shock protein", is well done, written and presented. It clearly deserves to be published in an appropriate journal, however, in my opinion, the content does not own the urgent character and originality for a publication in Communication Chemistry.

First, the presence of intermediate states in Csp proteins during mechanical unfolding has been already detected in other works, experimentally -as referenced in the manuscript- and theoretically (see also JPCL 2017 8, 5884, J. Phys. Chem. B 2018, 122, 11922).

Secondly, the fact that MT experiments are able to scan the low force regime is now well known (see PNAS 2019 116 (16) 787), and clearly present an advantage with respect to AFM.

Moreover, even if intermediated states are detected, no structural insights are provided in the manuscript. The general concept of melted globule used to rationalize the finding is quite generic.

Finally, it is also clear that the presence of intermediated states visible under the directional action of an external force might be not related to the folding / unfolding processes under physiological / functional conditions (e.g. temperature for Csp).

Reviewer #3 (Remarks to the Author):

This manuscript by Hong et al. deals with the single-molecule analysis of the mechanical unfolding of CspB using magnetic tweezers. Csp B is a protein that obeys a two-state folding mechanism in chemical denaturation experiments (both in bulk and using smFRET), but which has been shown to have a very complex mechanical unfolding process with multiple intermediates and parallel pathways using advanced AFM force spectroscopy and steered MD simulations. Using the stability of magnetic tweezers and the spy-catch tethering technology, the authors manage to make very long measurements of the CspB under force ramp or passive constant force conditions in which they manage to see stochastically alternating patterns of unfolding and folding events.

The use of CspB as model system for their characterization is a smart one, because it has been amply characterized before by alternative single-molecule methods, including force spectroscopy and smFRET, and therefore there is plenty of data to compare with. In this regard, the application of magnetic tweezers and the spy-catch tethering allows the authors to reliably operate in the low force regime (below pN), and more importantly, to measure in times sufficiently long as to resolve not only mechanical unfolding (done before by force extension AFM and then by force ramp and constant force AFM by other authors), but also folding, and the stochastic interconversions between the folded and unfolded states. The utilization of magnetic tweezers to study the single-molecule mechanical (un)folding of CspB is a smart choice. Moreover, the results are potentially sufficiently novel and complementary to the existing ones as to be of interest to the community of protein biophysicists and biochemists.

However, this interest and promise falls short in the current manuscript due to the lack of a serious comparison with the most relevant data on CspB, and on the most recent data on the observation of repeated mechanical folding-unfolding events of proteins by AFM and optical tweezers. I provide below a list of things that the authors need to satisfactorily address before publication so that their new data helps advance the field and does not create more confusion. In addition, there are also some style-formatting issues that need to be corrected. For instance, the authors put all the formulas they use for their analysis in supplementary information. However, they make constant references to those formulas, and the entire paper is about discussing the results obtained with the various formulas so that it is impossible to follow the work without constant referencing to the supplementary information. This is in my view a misuse of supplementary information, whose purpose is to supplement rather than provide the main body of the arguments put forward in the manuscript. I strongly recommend that the authors move all those formulas to the main text, so that the reader can assess what they are doing and describing without needing to constantly go to the supplementary information. They can keep the technical details of the analyses in sup. inf. but the formulas themselves, and the references to the original papers in which they are developed should be all given in the main manuscript.

Technical/scholarship issues that need addressing:

- 1) The authors need to do a more comprehensive and relevant comparative analysis of their results with prior works on this protein. In the current manuscript they focus almost exclusively on the first report of mechanical unfolding of CspB using force extension AFM. However, they need to compare their results much more thoroughly with the later force-ramp force-clamp AFM work, which they kind of misrepresent in their manuscript, and with the chemical denaturation data. For instance:
 - a. Their unfolding rate of 0.002 s^{-1} is too slow compared to bulk experiments (0.02 s^{-1} , Perl et al Nature Struct. Biol. 1998), and compared to the force clamp AFM (0.07 s^{-1} , Schonfelder Nature

Comms 2016). Why is their rate so much slower? This also make them overestimate the free energy of unfolding in the absence of force by 2.5 kBT).

b. The force-clamp AFM prior results report the observation of a large number of traces that are single steps: about 55% at 20 pN and 65% at 80 pN, whereas the maximum heterogeneity is at 40-60 pN (see figure 3a in the Nat. Comms. 2016 paper). Therefore, the authors are misrepresenting the prior work by saying it only observed intermediates. At least 50% of the traces were single steps as they report here. Given the trends with force in the Nat. Comms. report one would expect more single steps at the lower forces that the authors use here. They should discuss and acknowledge that. They could also test this possibility by performing experiments at higher constant force and at higher loading rates (their ramp experiments are all done at very low loading rates). It is great they can do the experiments at very load rates because they are more conservative (thanks to the extended time afforded by magnetic tweezers and spy-catch), but it would be important to replicate (or at least get close to) the conditions of the AFM experiment to see whether the multiple intermediates are indeed specific of an intermediate force range (20-80 pN) and can be replicated in their setup.

c. They should also discuss the relevance/significance of the superfast k_f they get from direct analysis ($6 \times 10^7 \text{ s}^{-1}$) that is at least 60-fold faster than folding speed limits.

d. They make a point about comparing their rates with the timescales of MD simulations in ns, but the steered MD simulations are done at much, much higher loading rates, and the timescales are not comparable to the experimental ones.

2) There is a significant body of work on the effects of the dynamics of the pulling device on the intrinsic molecular dynamics of the protein that is being studied by force spectroscopy that the authors do not mention, even though it is very significant for the interpretation and comparison of their data with those of others with other instruments. Particularly, the authors should comment on the theoretical analysis of instrumental dynamic effects by Cossio, Szabo and Hummer (PNAS 2015, JACS 2018).

3) The most relevant mechanical unfolding work (on other proteins) for this manuscript are the recent reports of stochastic unfolding-folding patterns measured by optical tweezers on the prion protein (Neupane Science 2016), and by AFM on the fast folding protein gpW (Schonfelder Comms Chem. 2018), as well the analysis of the strong dynamic effects observed in these experiments for both rates and transition paths using the Cossio et al analysis (Desancho, JPC 2018). All of these effects should be incorporated into their analysis of the folding unfolding rates from magnetic tweezers (data from figure 2).

Reviewers' comments:

Reviewer #1 (Remarks to the Author):

The work by Hong et al investigates the (un)folding of Csp from T maritime using magnetic tweezers. The authors investigate the unfolding and folding process over a range of forces and conclude that the protein is a two-state folder, in contrast to previous work using AFM. Magnetic tweezers surely have better resolution than AFM but I find one potential problem that the authors should comment on.

Response: Thanks for the reviewer's summary of our manuscript. We found that Csp mostly folds and unfolds with a single step, in contrast to previous force-clamp AFM experiment which observed long lifetime intermediate states with more than 50% probability. Because the unfolding rates show different force sensitivities at different forces ranges, which determines a free energy landscape with two barriers and a transient intermediate state.

Magnetic tweezers force measurement cannot be calibrated directly, unlike AFM, so it relies on indirect measurements of beads position. Perhaps the most relevant figure to conclude the two-state of Csp is Figure 3, in which there is a clear drift on the baseline force claimed to be 1 pN. Given this low value, a simple inspection reveals a drift of maybe 3-5 pN which completely perturbs the folding process of Csp. So, there is a real chance that the protein is not visiting such a low force to secure proper formation of all the contacts in the protein.

Response: The force of the magnetic tweezers is determined by the distance between the magnet and the bead. We measured the drifting velocity of paramagnetic beads M270 in 90% glycerol solution and calculated the forces by Stokes' law. In the Figure below, the force as a function of the magnet distance d can be fitted by an empirical equation:

$$F(d) = C_1 \left[\exp\left(-\frac{d}{r_1}\right) + C_2 \exp\left(-\frac{d}{r_2}\right) \right]$$

where $C_1, C_2, r_1,$ and r_2 are the fitting parameters. Detailed force calibration methods refer to our previous publication (Nanoscale, 2021,13, 11262-11269).

According to the formula for force calibration, if the baseline force drift from 1 pN to 2 pN, the distance will change more than 0.5 mm. The bi-directional repeatability of our translation stage (PI LS-110) is less than one micron. Therefore, the force is almost intrinsic constant with error of less than 0.0013 pN. The drift observed on extension (Fig. 3) is mainly due to the focus drift of the microscope. We have experimental data with very little extension drift and the same single step unfolding process. We replaced the data in Fig. 3 with new experimental data without drift of the baseline at 1 pN.

Also, why do they see about 20-30 nm previous extension after each pulling?

Response: We are not sure what “previous extension” refers to. We suppose that it is the 20-30 nm extension increase after the force jump from 1 pN to higher forces from 10 to 50 pN. This extension increase is due to the orientation alignment of the folded protein domains and the paramagnetic bead along the force direction.

In addition, they should show the full unfolding of the construct including the I27 modules. Even so, they should make a proper statistic with more than one single molecule, how many traces of single molecules recorded a varying force do they have?

To make a proper assumption about the folding/unfolding a good number of traces must be shown, because even in the AFM there is a chance to see single-event unfolding at different forces, in fact, this is the most probable scenario.

Response: Thanks for the suggestion. Fig. 1 shows the unfolding steps of I27 to identify the correct protein tether. In fact, each tether has been confirmed with I27 unfolding signal before the measurement. We give more examples with the I27 unfolding steps in the Supplementary Figure 2.

The number of independent protein tethers, and the number of folding and unfolding events in Fig.1c, 1d, 1e, 2b, 2c, 4a, 4b are all given in captions. Results of each individual tether are shown in Supplementary Figure 5 and its caption.

For constant loading rate measurements, the number of tethers is five, and the numbers of unfolding events at different loading rates are from 34 to 120. In the equilibrium measurements, we repeated the experiment on 4 different tethers and totally 116/115, 79/80, 210/212, 219/221, 66/67 unfolding/folding events of Csp at forces from 5 to 7 pN were collected. For the force-jump measurements, the same constant force experiments were repeated on 4 different tethers, and the unfolding rates were calculated from 89, 96, 98, 99, 96 unfolding events of Csp for forces 10 pN, 20 pN, 30 pN, 40 pN, and 50 pN, respectively. (see change list No. 14 and No. 16).

Reviewer #2 (Remarks to the Author):

Dear Editor,

The manuscript “Free energy landscape with two barriers and a transient intermediate state determining the unfolding and folding dynamics of cold shock protein”, is well done, written and presented. It clearly deserves to be published in an appropriate journal, however, in my opinion, the content does not own the urgent character and originality for a publication in Communication Chemistry.

Response: Thanks for the reviewer's summary of our manuscript. Our results demonstrate the importance of measurement over large force range. AFM force clamp

experiment relies on a feedback system to maintain a constant force. As feedback system usually possesses finite response time, it might lead errors in measurements. In magnetic tweezers, well-calibrated intrinsic constant forces are applied without any feedback system. And our work did the equilibrium measurement of folding and unfolding of Csp, especially the force-dependent folding rate which have never been measured before.

First, the presence of intermediate states in Csp proteins during mechanical unfolding has been already detected in other works, experimentally -as referenced in the manuscript- and theoretically (see also JPCL 2017 8, 5884, J. Phys. Chem. B 2018, 122, 11922).

Secondly, the fact that MT experiments are able to scan the low force regime is now well known (see PNAS 2019 116 (16) 787), and clearly present an advantage with respect to AFM.

Moreover, even if intermediated states are detected, no structural insights are provided in the manuscript. The general concept of melted globule used to rationalize the finding is quite generic.

Response: The authors studied the mechanical stability of Csp by computational simulation and observed unfolding intermediate states in JPCL 2017, 8, 5884 and J. Phys. Chem. B 2018, 122, 11922. In MD simulation, it is expectable to observe multiple intermediate states due to its almost unlimited time resolution and spatial resolution.

Thanks to the advantages of MT, we have the capability to measure the force-dependent folding and unfolding rates over force range beyond the scope of AFM. We demonstrate that the measurement over large force range is necessary to disclose the full free energy landscape of proteins.

The long-lifetime intermediate states observed by force-clamped AFM have lifetime of seconds, which were not detected by our MT experiments. With the wide force-range (5-50 pN) of force-dependent folding/unfolding rates, we indirectly detected the intermediate state between the two transition states TS1 and TS2. Its life time is shorter than the time resolution of our magnetic tweezers, 5 ms. Therefore, it is

different from the intermediate states observed in AFM which has lifetime of several seconds. The possible candidate of the short-lifetime intermediate state in our model is the theoretically predicted molten globule state. Because it is a general intermediate state for globular proteins, it might not have specific structure, as it is supposed to be a relative compact structure with partially formed secondary structures.

Finally, it is also clear that the presence of intermediated states visible under the directional action of an external force might be not related to the folding / unfolding processes under physiological / functional conditions (e.g. temperature for Csp).

Response: We admit that this point is always a major concern of force spectroscopy experiments. Our magnetic tweezers measurements at low forces down to 5 pN makes the distortion of free energy landscape by force much smaller than previous experiment done at higher forces. Therefore, the extrapolation from low force results to zero-force condition is more reliable. Combination of force spectroscopy with controlling of biochemical environmental conditions (temperature, denaturant, etc.) should be able to provide more information, which is out of the scope of current paper.

Reviewer #3 (Remarks to the Author):

This manuscript by Hong et al. deals with the single-molecule analysis of the mechanical unfolding of CspB using magnetic tweezers. Csp B is a protein that obeys a two-state folding mechanism in chemical denaturation experiments (both in bulk and using smFRET), but which has been shown to have a very complex mechanical unfolding process with multiple intermediates and parallel pathways using advanced AFM force spectroscopy and steered MD simulations. Using the stability of magnetic tweezers and the spy-catch tethering technology, the authors manage to make very long measurements of the CspB under force ramp or passive constant force conditions in which they manage to see stochastically alternating patterns of unfolding and folding events.

Response: Thanks for the reviewer's excellent summary of our manuscript.

The use of CspB as model system for their characterization is a smart one, because it has been amply characterized before by alternative single-molecule methods, including force spectroscopy and smFRET, and therefore there is plenty of data to compare with. In this regard, the application of magnetic tweezers and the spy-catch tethering allows the authors to reliably operate in the low force regime (below pN), and more importantly, to measure in times sufficiently long as to resolve not only mechanical unfolding (done before by force extension AFM and then by force ramp and constant force AFM by other authors), but also folding, and the stochastic interconversions between the folded and unfolded states. The utilization of magnetic tweezers to study the single-molecule mechanical (un)folding of CspB is a smart choice. Moreover, the results are potentially sufficiently novel and complementary to the existing ones as to be of interest to the community of protein biophysicists and biochemists.

Response: Thanks for the reviewer's positive comments to our work.

However, this interest and promise falls short in the current manuscript due to the lack of a serious comparison with the most relevant data on CspB, and on the most recent data on the observation of repeated mechanical folding-unfolding events of proteins by AFM and optical tweezers. I provide below a list of things that the authors need to satisfactorily address before publication so that their new data helps advance the field and does not create more confusion.

Response: Thanks for the reviewer's constructive suggestion. We compared our results with previous work on Csp proteins. The details of responses and revisions are given below point to point.

In addition, there are also some style-formatting issues that need to be corrected. For instance, the authors put all the formulas they use for their analysis in supplementary information. However, they make constant references to those formulas, and the entire paper is about discussing the results obtained with the various formulas so that it is impossible to follow the work without constant referencing to the supplementary information. This is in my view a misuse of supplementary information, whose purpose

is to supplement rather than provide the main body of the arguments put forward in the manuscript. I strongly recommend that the authors move all those formulas to the main text, so that the reader can assess what they are doing and describing without needing to constantly go to the supplementary information. They can keep the technical details of the analyses in sup. inf. but the formulas themselves, and the references to the original papers in which they are developed should be all given in the main manuscript.

Response: Thanks for the reviewer's constructive suggestion. We have moved the equations used to analyze or fit the experimental results to the main text: section "Materials and Methods" – subsection "Theoretical model and data analysis". (see change list No. 13)

Technical/scholarship issues that need addressing:

1) The authors need to do a more comprehensive and relevant comparative analysis of their results with prior works on this protein. In the current manuscript they focus almost exclusively on the first report of mechanical unfolding of CspB using force extension AFM. However, they need to compare their results much more thoroughly with the later force-ramp force-clamp AFM work, which they kind of misrepresent in their manuscript, and with the chemical denaturation data. For instance:

a. Their unfolding rate of 0.002 s^{-1} is too slow compared to bulk experiments (0.02 s^{-1} , Perl et al Nature Struct. Biol. 1998), and compared to the force clamp AFM (0.07 s^{-1} , Schonfelder Nature Comms 2016). Why is their rate so much slower? This also make them overestimate the free energy of unfolding in the absence of force by $2.5 \text{ k}_B\text{T}$.

Response: Thanks for pointing out this major difference. We did not emphasize this point in the original manuscript. In fact, this is one of our major founding from low-force measurement by magnetic tweezers.

The biochemistry bulk experiment and AFM experiment use experimental results obtained from high concentration of denaturant and high force to extrapolate to conditions with zero concentration of denaturant and zero force, respectively. There is an assumption that the linear extrapolation is valid, but we found that this is not the case (Fig. 4).

To compare our experimental results with previous AFM experiments, we did new experiments with higher loading rates 8 pN s⁻¹ and 16 pN s⁻¹. We found the fitting results are highly dependent on the range of loading rates (revised Fig. 1e). At loading rates greater than 4 pN s⁻¹, the linear fitting by Bell's model gives $k_u^0 = 0.03 \text{ s}^{-1}$, which is consistent with the bulk experiments and previous AFM result. However, at smaller loading rates, both unfolding force distribution fitting (Fig. 1c) and linear fitting of different loading rates (Fig. 1e) give smaller k_u^0 and bigger x_u .

Additionally, the force-dependent unfolding rate obtained by constant force measurement also give the same results: fitting at forces greater than 10 pN gives $k_u^0 = 0.032 \text{ s}^{-1}$, while fitting at forces smaller than 7 pN gives $k_u^0 = 0.0002 \text{ s}^{-1}$.

In the revised manuscript, we emphasized this point. (see change list No. 8, No. 14 and No. 16)

b. The force-clamp AFM prior results report the observation of a large number of traces that are single steps: about 55% at 20 pN and 65% at 80 pN, whereas the maximum heterogeneity is at 40-60 pN (see figure 3a in the Nat. Comms. 2016 paper). Therefore, the authors are misrepresenting the prior work by saying it only observed intermediates. At least 50% of the traces were single steps as they report here. Given the trends with force in the Nat. Comms. report one would expect more single steps at the lower forces that the authors use here. They should discuss and acknowledge that.

They could also test this possibility by performing experiments at higher constant force and at higher loading rates (their ramp experiments are all done at very low loading rates). It is great they can do the experiments at very load rates because they are more conservative (thanks to the extended time afforded by magnetic tweezers and spy-catch), but it would be important to replicate (or at least get close to) the conditions of the AFM experiment to see whether the multiple intermediates are indeed specific of an intermediate force range (20-80 pN) and can be replicated in their setup.

Response: Thanks for pointing out our incorrect description of the results by force-clamp AFM. We have corrected the description in the main text. (see change list

No. 1 and No. 12)

In our constant-force experiment of 10-50 pN which covers more than 50% of the force range of AFM (20-80 pN), all the unfolding processes are single-step transitions without long-lifetime intermediate state if the measurement is finished within three hours. Due to the limitations of the time resolution (5 ms time step and 0.15 s to move the magnets) of our equipment, we can hardly observe unfolding steps when force is bigger than 50 pN (Supplementary Figure 11). The Csp protein will unfold within 0.15 s before the force increased to the setting values of 60 pN or 80 pN. So, we did not add new unfolding data points at constant forces higher than 50 pN.

We did new experiment with the higher loading rate of 8 pN s⁻¹ and 16 pN s⁻¹, the unfolding of Csp is still single-step transition (Fig. 1 and Supplementary Figure 1). (see change list No. 5 and No. 14)

c. They should also discuss the relevance/significance of the superfast k_f they get from direct analysis ($6 \times 10^7 \text{ s}^{-1}$) that is at least 60-fold faster than folding speed limits.

Response: Sorry for the confusion caused by the fitting results in Fig. 1d. This ultrafast folding rate k_f^0 of Csp is obtained by fitting with Bell's model, which is unsuitable to describe the force-dependent folding rate. We have marked clearly in Fig. 1d to indicate that the wrong fitting results is due to the improper model. Fig. 4 a and b give the zero-force folding rate of 400 s^{-1} which is consistent with previous biochemistry results. (see change list No. 14 and No. 16)

d. They make a point about comparing their rates with the timescales of MD simulations in ns, but the steered MD simulations are done at much, much higher loading rates, and the timescales are not comparable to the experimental ones.

Response: Thanks for pointing out this confusing timescale problem. We did not try to compare our experimental results of unfolding rates with that of MD simulation. We agree with the reviewer that the rates from MD and experiments are difficult to

compare with each other directly, because the full atomic MD are done at 200 pN, much higher than the experimental force range. We revised the sentence which describe the relationship between AFM experiments and simulation. (see change list No. 2)

2) There is a significant body of work on the effects of the dynamics of the pulling device on the intrinsic molecular dynamics of the protein that is being studied by force spectroscopy that the authors do not mention, even though it is very significant for the interpretation and comparison of their data with those of others with other instruments. Particularly, the authors should comment on the theoretical analysis of instrumental dynamic effects by Cossio, Szabo and Hummer (PNAS 2015, JACS 2018).

Response: Thanks for introducing these two important papers. Effect of apparatus used in single molecule manipulation experiments are often very important to interpret the results and compare experiments with different techniques. Due to the limited time resolution in magnetic tweezers (5 ms in our experiments), the transition path time (usually in microsecond time scale) cannot be recorded. Comments on the instrumental dynamic effects were added in discussion section (see change list No. 11).

3) The most relevant mechanical unfolding work (on other proteins) for this manuscript are the recent reports of stochastic unfolding-folding patterns measured by optical tweezers on the prion protein (Neupane Science 2016), and by AFM on the fast folding protein gpW (Schonfelder Comms Chem. 2018), as well the analysis of the strong dynamic effects observed in these experiments for both rates and transition paths using the Cossio et al analysis (Desancho, JPC 2018). All of these effects should be incorporated into their analysis of the folding unfolding rates from magnetic tweezers (data from figure 2).

Response: Thanks for introducing these three relevant papers. Based on the theory of Cossio et al and its application in the paper of De Sancho, JPCB 2018, we estimated the apparatus transition rate k_A to be about 620 s^{-1} , which is more than one order of magnitude bigger than the fastest unfolding rate of Csp at 50 pN in our experiments.

Therefore, the effect of slow apparatus rate can be neglected in our measurement. Estimation of the apparatus rate was added in Supplementary Note 2 and Figure 9, and comments were added in discussion section (see change list No. 11).

Main change list

Main changes in the revised manuscript are listed below. All main changes are marked with red color in main text and supplementary information. The new experiment results and statistics have been updated and marked with red color in main text.

Main text:

1. Page 2, paragraph 3: Description of AFM experimental results was updated to be more precise: “Csp was found to unfold with about 50% probability through multiple heterogeneous pathways with long-lived intermediate states (lifetime of seconds) at moderate force range from 40 pN to 60 pN.”
2. Page 3, paragraph 1: Discussion of the MD simulation results was updated to “Though the simulations support the existence of unfolding intermediate states, the lifetime of the intermediate states cannot be compared directly between experiments and simulations.”
3. Page 4, paragraph 2: A sentence was added: “ $k_u(f) = k_u^0 \exp(f x_u / k_B T)$, where k_u^0 denotes the unfolding rate at zero force if the formula can be extrapolated to zero force, x_u the unfolding distance, k_B the Boltzmann constant, and T the absolute temperature.”
4. Page 4, paragraph 2: Add a sentence to discuss the fitting results of k_f^0 . “This k_f^0 is even faster than folding speed limit, which is clearly an unreasonable result from improper model.”
5. Page 4, paragraph 2: Replaced the constant loading rates fitting results of Fig 1e with “Fitting with Eq. (3) gives $k_u^0 = 0.002 \text{ s}^{-1}$ and $x_u = 1.42 \text{ nm}$ at loading rates from 0.4 pN s^{-1} to 4 pN s^{-1} , and gives $k_u^0 = 0.03 \text{ s}^{-1}$ and $x_u = 0.64 \text{ nm}$ at loading rates from 4 pN s^{-1} to 16 pN s^{-1} .”

6. Page 6, paragraph 1: Add a sentence to explain the signal after Csp unfolding event at 30 pN in updated Fig. 3: “The signal after Csp unfolding at 30 pN is the unzipping/zipping dynamics of N-SpyTag/SpyCatcher in the tether. This signal can serve as the fingerprint to identify the correct protein tether.”
7. Page 6, paragraph 2: The sentence to compare our result with previous publications was revised by adding: “and biochemistry study which rely on linear extrapolation from unfolding rates measured at high force or high concentration of denaturant conditions, respectively”
8. Page 6, paragraph 2: Revised the sentence to compare the unfolding rate from different force ranges by adding: “which is more than two orders of magnitude smaller than $k_{u,1}^0$ ”
9. Page 7, paragraph 1: Add a clause to compare our result to others. “which is consistent with previous biochemistry study.”
10. Page 7, paragraph 2: Add a clause to compare our results with previous measurements: “which is slightly larger than previous biochemistry measurements (8.5-10.5 $k_B T$).”
11. Page 8, paragraph 2: Add a paragraph to discuss whether the experimental results measured by MT need to be corrected.
12. Page 8, paragraph 4: Gave more precise description of previous AFM experiments by updated clause: “which detected several long-lifetime intermediate states during the unfolding process at 40 and 60 pN with more than 50% probability.”
13. Page 11, paragraph 1: Add a subsection “Theoretical model and data analysis”, which is moved from supplementary information of original manuscript.

Figures in main text:

14. Page 13, Fig. 1, Fig. 1d: Fitting parameters were updated and “Unreasonable k_f^0 from Bell’s model fitting!” was marked. Fig. 1e: New experimental data were added, and fitting was done in different range of loading rates. The figure caption is revised accordingly.

15. Page 15, Fig. 3: Replaced the figure with new experimental data, and figure caption was updated accordingly.
16. Page 16, Fig. 4: Updated the average data from four tethers, the fitting results, and figure caption.

Supplementary Information:

17. Supplementary Page 2: Supplementary Table 1 and Figure 8 were updated.
18. Supplementary Page 3: Supplementary Note 2 and Figure 9 were added to explain the calculation process of apparatus rate k_A .
19. Supplementary Page 4: Supplementary Figure 1 was added to show the different constant loading rates measurements of Csp.
20. Supplementary Page 5: Supplementary Figure 2 was added to show I27 unfolding steps.
21. Supplementary Page 7: Supplementary Figure 4 was added to show unfolding of Csp at force-jump measurement.
22. Supplementary Page 8: Supplementary Figure 5 was added to show force-dependent folding rates and unfolding rates of Csp from four individual tethers and the average results.
23. Supplementary Page 12: Supplementary Figure 9 was added to show free energy as a function of extension obtained from the extension distribution at 6.3 pN.
24. Supplementary Page 14: Supplementary Figure 11 was added to show the time for magnets to arrive at the setting distance.

REVIEWERS' COMMENTS:

Reviewer #1 (Remarks to the Author):

The authors should be aware that they have a technical limitation in time resolution that makes difficult any comparison with previous work by AFM. they should acknowledge that. Other than that I think the work is good.

Reviewer #3 (Remarks to the Author):

The authors have addressed most of the comments/requests I made. The revised manuscript might be publishable in Comms. Chem.

REVIEWERS' COMMENTS:

Reviewer #1 (Remarks to the Author):

The authors should be aware that they have a technical limitation in time resolution that makes difficult any comparison with previous work by AFM. they should acknowledge that. Other than that I think the work is good.

Response: Thanks. we added one sentence in the third paragraph in section "Discussion" to address that the proposed transient intermediate was not recorded directly by magnetic tweezers due to the limited five-millisecond time resolution of our magnetic tweezers.

Reviewer #3 (Remarks to the Author):

The authors have addressed most of the comments/requests I made. The revised manuscript might be publishable in Comms. Chem.

Response: Thanks.